# To eliminate trachoma: Azithromycin mass drug administration coverage and associated factors among adults in Goro district, Southeast Ethiopia

**Tadele Feyisa[1], Desalegn Bekele[2], Birhanu Tura[2], Ahmednur Adem[2], Fikadu Nugusu[ID][2]***

**1** Robe town Health Office, Bale zone of Oromia regional state Health Office, Robe, Ethiopia, **2** Department of Public Health, School of Health Sciences, Madda Walabu University, Goba, Ethiopia

* fikadu215@gmail.com

**Data Availability Statement:** All relevant data are within this manuscript and its Supporting information file.

## Abstract

### Background

Globally, although effective prevention strategies and treatment are available, trachoma remains the major cause of infectious loss of sight. Trachoma is a predominant neglected disease in Ethiopia, and there is a 40.4% prevalence of active trachoma in the Goro district, Southeast Ethiopia. World Health Organization (WHO) recommends azithromycin mass treatment of at least 80% coverage to eliminate trachoma, even though the coverage of azithromycin mass treatment has not been studied yet in depth. Thus, this study aimed to assess the coverage and factors influencing azithromycin mass treatment uptake among adults in Goro district, Southeast Ethiopia.

### Methods

A community-based cross-sectional study was conducted from April 1st to April 30th, 2021 among all adults aged 15 years old and above. The multistage sampling technique was used to select 593 study respondents. A structured interviewer-administered questionnaire was used. Data were entered into Epi-Data version 3.1 and analyzed using SPSS version 23.0 software. Descriptive analysis and binary logistic regression analysis were used to analyze the data. Adjusted odds ratios (AOR) along with a 95% confidence interval (CI) and p-value < 0.05 were used to declare the strength and the significance of association, respectively.

### Results/Principal findings

Five hundred and seventy eight study participants with a 97% response rate were included. The proportion of azithromycin mass drug administration coverage was found to be 75.80%; 95% CI: (72%-79%) in this study. Having better knowledge about trachoma (AOR = 2.36; 95% CI: 1.19–4.70), having better knowledge about azithromycin mass treatment (AOR = 4.19; 95% CI: 2.19–7.98), being educated (AOR = 7.20; 95% CI: 1.02–51.09), a campaign conducted at the quiet time (off-harvesting/planting season) (AOR = 6.23; 95% CI: 3.23–11.98), heard about the serious adverse effect from others (AOR = 0.25; 95% CI: 0.10–

**Funding:** The author(s) received no specific funding for this work.

**Competing interests:** The authors have declared that no competing interests exist.

0.59) and being a volunteer to take azithromycin in the next campaign (AOR = 5.46; 95% CI: 2.76–10.79) were significantly associated with azithromycin mass drug administration coverage.

## Conclusions/Significance

The proportion of azithromycin mass treatment coverage of this study was lower than the WHO minimum target coverage. Thus, strengthening awareness, enhancing azithromycin mass trachoma treatment messages, and conducting campaigns off-season outside of harvesting and planting time should be prioritized in the future to meet the 2030 Sustainable Development Goal (SDG) target.

### Author summary

Trachoma is the leading cause of infectious loss of sight worldwide. The mass treatment with azithromycin has been used in the prevention and treatment of chlamydia trachomatis and is recognized as a potentially vital and important public health strategy for the control of trachoma disease and other co-infections. Oral azithromycin is easy to administer and offers a better resolution for controlling blinding trachoma for long periods. The proportion of azithromycin mass treatment coverage in the Goro district was lower than the WHO minimum target. Having better knowledge about the trachoma and azithromycin mass treatment and being educated were some of the factors positively affecting the uptake of the azithromycin mass treatment. Hearing serious adverse effects of the drug from other people and campaigns conducted during harvesting or seeding time were some factors negatively affecting the coverage of azithromycin mass treatment in the current study.

## Introduction

Trachoma is the result of ocular infection with the bacterium chlamydia trachomatis and it is the leading cause of blindness worldwide, which is spread by direct contact with eye and nose discharges from infected individuals or by contacts with fomites (inanimate objects that carry infectious agents) like shared clothes and towels and by eye-seeking flies [1].

Globally, 232 million people are estimated to be living in trachoma-endemic areas; of those, 77% live in Africa [2]. Africa is the largest attacked continent with about 27.8 million (68.5%) cases of active trachoma and 3.8 million (46.6.%) cases of blinding trachoma of global share [3]. Ethiopia has the global highest burden of trachoma with 76.2 million people at risk of trachoma being within endemic areas and the national prevalence of active trachoma and trichiasis is 40.1% and 3.1% respectively [4,5].

In 1997 the World Health Organization (WHO) established an Alliance for Global Elimination of Trachoma by the year 2020 and it endorsed an integrated package of interventions called 'SAFE', which stands for, "surgery for trichiasis, antibiotic for ocular trachomatis infection, facial cleanliness and environmental improvement focusing on access to water and sanitation [6]. Although effective prevention strategies and treatment are available, trachoma remains the major cause of infectious blindness worldwide [7]. Despite considerable progress,

the target of eliminating trachoma globally by December 2020 was not achieved, and why its target date is 2030 in line with Sustainable Development Goals (SDGs) agenda [8].

Different treatment modalities were practiced for treating and preventing trachoma in different periods. In recent years, many randomized control trial studies have indicated that azithromycin is the treatment of choice for ocular infection with chlamydia trachomatis [9]. Azithromycin clears ocular chlamydia trachomatis infection with one oral dose of one gram or four tablets, 250 milligrams each for adults, and it is well-tolerated and reduces transmission in the community by treating the pool of infection [1]. However, elimination of active trachoma is only achieved as long as the azithromycin mass treatments (AMT) were given frequently enough and at a high enough coverage to greater than 80% on the basis that high treatment coverage will reduce re-emergence of infection in mass treated communities [10]. Improved coverage of AMT to treat trachoma is critical to global, national, and district success in the elimination of blinding trachoma [11]. However, full acceptance of the community for mass drugs administration has many challenges (still not meeting WHO target) due to different reasons; lack of information about the campaign, fear of side effects and being absent during the AMT campaign, and misconceptions about trachoma [12,13] resulting in re-infection [14]. One potential source of reinfection could even be individuals from a community who did not participate in mass treatment [15].

Ethiopia launched the 'VISION 2020' initiative in 2002 to ensure the best possible vision for all people, and the mass drug administration program started in 2003 [16]. In Ethiopia, the previous studies in the Awi zone in northwestern Ethiopia [12]; in northern Ethiopia [17], and in six districts of the Oromia region [13] only included household heads; who did not represent the other family members and had some unavoidable information disparities. Therefore, in the current study, these gaps were addressed by including all adults aged 15 years old and above in households. Furthermore, in Ethiopia, studies that addressed community uptake of azithromycin mass drug administration and related factors are very limited.

Oromia regional state is one of the most trachoma endemic regions in Ethiopia with an active trachoma prevalence of 41.3% indicating that trachoma is a significant public health problem in at least 218 districts of the region [4,16]. In the Goro district, active trachoma and trichiasis prevalence were 40.4% and 1.5% respectively [16]. According to Goro district health office azithromycin mass treatment annual campaign was first initiated in 2016. However, there is still a community resistance to taking azithromycin mass treatment during the campaign [18]. Hence, this study assessed the uptake of azithromycin mass drug administration and associated factors among adults in the Goro district, southeast Ethiopia.

## Methods

### Ethics statement

This study followed the Helsinki Declaration protocol. Ethical clearance was obtained from Madda Walabu University Institutional Review Board. An official permission letter was secured from the Goro district health office. Informed verbal and written consent of the respondents were obtained. For those study participants 15 to 18 years old, verbal consent was obtained from their families besides the assent obtained from each study participant. The confidentiality of information collected is kept by avoiding personal identifiers.

### Study design, period, and setting

A community-based cross-sectional study design was conducted from April 1st to 30th, 2021, among all adults aged 15 years old and above who resided in sampled households in Goro district, Bale zone, Southeast Ethiopia. Goro district is one of the 12 districts in the Bale zone of

the Oromia region, located in the southeast direction, 60km from zonal town, Robe, and 490km from the capital city of Ethiopia, Addis Ababa. The total population of the district is 119,713 with 61,054 males and 58,659 females within a total household of 24,940.

There are 56,995 adults aged 15 years and above residing in the district and the majority (84%) of the population live in the rural area; the remaining (16%) are living in the urban area. The district has one hospital, six health centers, and 23 health posts with 231 personnel working in health sectors with different health professionals; 47 rural Health Extension Workers (HEWs), four urban HEWs, and 85 administrative staff. Four rounds of azithromycin mass drug administration campaigns were conducted by HEWs, and health professionals in the Goro district, and the recent (fourth) campaign was conducted in December 2019 [18].

## Population

**Source population.**   All adult population aged 15 years and above living in Goro district, Bale zone, Southeast Ethiopia.

**Study population.**   All selected adult population aged 15 years and above living in Goro district, Bale zone, Southeast Ethiopia.

**Inclusion and exclusion criteria's.**   Adults who were critically ill and unable to communicate during data collection and non-permanent residents were excluded from the study.

## Sample size determination

A single population proportion formula of n = $(Z_{a/2})^2$ p (1-p) /$d^2$ was used to determine the sample size, with the assumption of; 1.96 (for 95% confidence interval, precision at d = 0.05 margin of error, where "n" is the required sample size, "p" is the proportion of azithromycin uptake 62.8% taken from a study done at Awi zone of Amhara regional state [12]. The ultimate sample size was 593 after accounting for a 10% predicted non-response rate and a design effect of 1.5.

## Sampling techniques

A multi-stage stratified sampling technique was employed. Initially, the stratification of kebeles into rural and urban strata was made. A total of Nine (One urban and Eight rural) kebeles *(the smallest admiration unit in the district)* were selected from the total kebeles in the district. Selections of kebeles were undertaken by simple random sampling. The sample size was proportionally allocated to each sampled kebeles to estimate the total number of adults. Finally, the respondents were selected by a systematic random sampling method, every 14th interval based on the existing list of households' from the family folder of the Community Health Information System (CHIS) as a sampling frame. In the case of more than one adult aged 15 years old and above in the selected household, one adult was selected using the lottery method for the interview.

## Study variables

**Dependent variable.**   Azithromycin Mass Treatment uptake.

**Independent variables.**   The socio-demographic and economic characteristics variables were:—age, sex, marital status, role in the family, occupation, ethnicity, religion, educational status, place of residence, and monthly income.

The knowledge variables about trachoma and AMT were heard about trachoma, cause of trachoma, ways of transmissions, trachoma prevention methods, how we treat it, heard about the AMT, importance of AMT, how AMT was given, AMT is free or charged and drug-related

variables were heard side effect of the drug, the experience side effect of the drug, ever been taken the drug, willingness to take in the future, the reason for not take and also campaign-related variables were timing and social mobilization.

## Measurements and operational definitions

**Adults.**   An adult aged 15 years old and above in this study [1].

**Azithromycin Mass Drug Administration (AMDA).**   It is an annual mass administration of azithromycin 1gram or 4 tablets each 250 milligram for all adults aged 15 years and above irrespective of disease status for trachoma elimination purposes [13,17].

**Coverage of azithromycin mass drug administration.**   Defined as self-reporting of azithromycin mass treatment of the last campaign by respondents. An adult who received azithromycin during the campaign was categorized as taken (coded 1) and those who were not received were categorized as not taken (coded 0) [12,13].

**Knowledge.**   Respondents were asked a total of nine knowledge assessment questions. The mean of the response was the cut point and was assessed by asking serial of questions related to trachoma disease and AMDA, correct responses were given a value of "1" and incorrect responses were given a value of "0" and those who scored greater than the mean were considered to have better knowledge.

## Data collection instruments

The interviewer-administered structured questionnaire was used to collect data from each study subject, which was adopted from International Trachoma Initiative guidelines and different works of literature [12,13,17]. The interview was conducted in their respective household. The data collector checked the questionnaire for completeness before leaving each respondent. Revisit was done on the next day for those who were not present at home during data collection.

## Data quality controls

The questionnaire was prepared initially in English language and then translated into a local language, Afan Oromo, by a language expert, and back-translated to English. Three days intensive training was given for data collectors and supervisors on the objective of the study and each component of the questionnaire. One week before the actual data collection period, a pre-test was conducted on 5% (30 adults) of the sample from a similar population outside the study area in Gamora kebele, and based on the findings of the pre-test, minor modifications of questions, wordings, phrases like AMT and campaign-related variables and time required to interview respondents' modifications were conducted.

Each questionnaire collected from the field was checked for completeness, missed values, and unlikely responses, and manually cleaned. Also, to minimize recall bias, an oval-shaped and bright pink-colored azithromycin sample tablets were shown for each respondent and Coronavirus Disease 2019 (COVID-19) protection practice was under consideration. Supervisors and investigators closely monitored the data collection process. Data were cross-checked for consistency and accuracy.

## Data processing and analysis

Data were coded and entered to EpiData version 3.1 and exported to SPSS version 23.0 computer software packages for analysis. The binary logistic regression model was fitted to assess factors associated with uptake of azithromycin. The model's fitness was evaluated using the

Hosmer and Lemeshow goodness of fit test at the p-value > 0.05. All independent variables that had a p-value <0.25 in the bivariate logistic regression analysis were entered for the multi-variable logistics regression model to control for confounding variables. Multi-collinearity was checked using variance inflation factor (VIF) at a value <10. Independent variables with a p-value < 0.05 in the multivariable logistic regression analysis were considered to have a statistically significant association with the azithromycin mass treatment coverage. Descriptive statistics; frequencies, percentages, and mean were used to report the findings. The degree of association was declared using Adjusted Odds Ratios (AOR) along with a 95% confidence interval (CI).

## Results

### Socio-demographic and socio-economic characteristics of the respondents

In this study, of 593 respondents 578 have participated with a response rate of 97%. Majority of the study participants 507 (87.70%) were rural residents and half 290 (50.20%) of respondents were females. The mean age of the respondents was 35.05 ± 13.75. The majority of the respondents, 248 (42.90%) were farmers in occupation. Majority of the respondents, 469 (81.10%) were followers of Muslims and 559 (96.70%) were from Oromo ethnic group. Regarding educational status, 287(49.60%) did not have formal education and 187 (32.40%) attended primary education. The majority of the respondents (69.80%) earned less than 1000 Ethiopian Birr (ETB) monthly income (Table 1).

### Coverage of Azithromycin mass treatment

The proportion of azithromycin mass drug administration coverage in this study was found to be 75.80% (95% CI; [72%, 79%]) (Fig 1).

The uptake status of the urban and rural residents from this study was 77.50% (95% CI; [70%, 89%]) and 75.50% (95% CI; [75%, 83%]), respectively. Accordingly, the azithromycin mass drug administration uptake status of male and female respondents was 69.40% (95% CI; [68%, 78%]) and 82.10% (95% CI; [81%, 89%]) respectively. One-fourth of the respondents, 140 (24.20%) had not taken azithromycin during the mass treatment campaign due to different reasons (Table 2).

Out of the total four rounds of azithromycin mass treatment campaign conducted in the district, almost half, 283(48.90%) of the respondents participated in the campaign and received the azithromycin more than two times. However, the number of respondents who had participated and received the azithromycin mass treatment four times was only 114 (19.70%). On average, each respondent had taken azithromycin mass treatment 2.69 times in the past four rounds of the campaign (Fig 2).

Among the total respondents who participated in the study, three-fourths of them, 441 (76.30%) responded that they are willing to take azithromycin in the next campaign if it continues in the future. Accordingly, nearly one-fourth 137(23.70%) of the respondents would not volunteer to receive azithromycin in the coming campaign mentioning the different reasons for withdrawal from the future AMT campaign (Fig 3).

### Trachoma related

Majority of the respondents, 506 (87.50%) had ever heard about trachoma and the most source of information 405 (80%) was from health workers. More than three-fourths of the respondents, 451 (78%) and 453(78.40%) knew what causes trachoma and realized transmission ways, respectively. Of those who responded that trachoma is transmitted from person to

**Table 1. Socio-demographic characteristics of the respondents in Goro district, Bale zone, Southeast Ethiopia, 2021 (n = 578).**

| Variable | Category | Frequency | Percentage |
|---|---|---|---|
| **Sex** | | | |
| | Male | 288 | 49.80 |
| | Female | 290 | 50.20 |
| **Age** | | | |
| | 15–24 | 146 | 25.30 |
| | 25–34 | 177 | 30.60 |
| | 35–44 | 115 | 19.90 |
| | 45–54 | 85 | 14.70 |
| | > = 55 | 55 | 9.50 |
| **Occupation** | | | |
| | Farmer | 248 | 42.90 |
| | House Wife | 160 | 27.70 |
| | Student | 69 | 11.90 |
| | Government Worker | 45 | 7.90 |
| | Merchant | 36 | 6.70 |
| | Unemployed | 17 | 2.90 |
| **Educational status** | | | |
| | Not formal education | 287 | 49.60 |
| | Primary education | 187 | 32.40 |
| | Secondary education | 53 | 9.20 |
| | College & above | 51 | 8.80 |
| **Residence** | | | |
| | Rural | 507 | 87.70 |
| | Urban | 71 | 12.30 |
| **Marital status** | | | |
| | Single | 128 | 22.10 |
| | Married | 404 | 69.90 |
| | Divorced | 19 | 3.30 |
| | Widowed | 27 | 4.70 |
| **Monthly income (ETB)** | | | |
| | 100–1000 | 403 | 69.80 |
| | 1001–4000 | 116 | 20 |
| | >4000 | 59 | 10.20 |
| **Religion** | | | |
| | Muslim | 469 | 81.10 |
| | Orthodox | 94 | 16.30 |
| | Protestant | 15 | 2.60 |
| **Ethnicity** | | | |
| | Oromo | 559 | 96.70 |
| | Amhara | 19 | 3.30 |

person, 391(67.60%) frequently mentioned the fly as a transmission modality. Concerning trachoma prevention methods, 434(75.10%) responded that trachoma is a preventable disease, and 252(43.60%) mentioned taking azithromycin drug for trachoma treatment. The mean knowledge score of respondents was 4.08 ± 1.64 standard deviation (SD) and the majority 447 (77%) of respondents scored above the mean score and classified as having a better knowledge of trachoma (Fig 4).

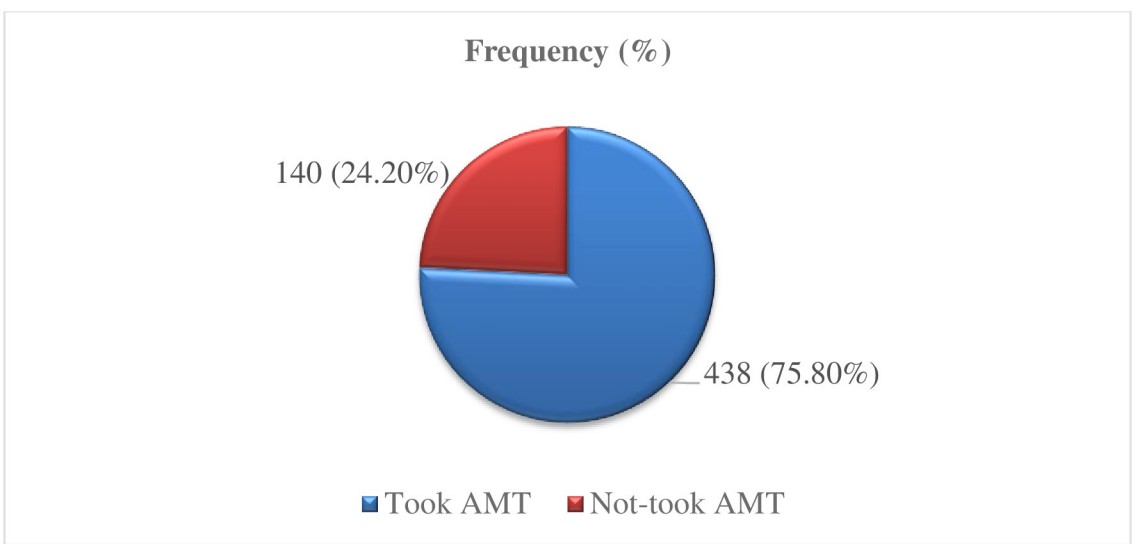

**Fig 1. Coverage of azithromycin mass treatment of respondents in Goro district, Bale zone, Southeast Ethiopia, 2021.**

## Azithromycin mass treatment

Majority of the respondents, 507(87.70%) knew AMT drugs were given for free with no payment. The mean knowledge score of respondents on AMT was 3.16 ± 1.12 standard deviation (SD). More than half, 327(57%) of respondents scored above the mean score and classified as having a better knowledge of azithromycin mass treatment.

Among the respondents who ever heard about azithromycin mass treatment, 507(87.70%) most of them heard from health workers (Fig 5).

## Prevalence of reported side effects and serious adverse effects

Out of the total, 167(28.90%) respondents had ever encountered side effects related to azithromycin during mass treatment (Fig 6).

Among the total respondents who participated in the study, 208 (36%) had ever heard other persons complaining about the side effects after they took the drug. Out of these, 77 (37%) had ever heard other persons complaining about the serious adverse effects of azithromycin.

**Table 2. Response to question related to the reasons for not-took azithromycin mass treatment campaign by the respondents in Goro district, Bale zone, Southeast Ethiopia, 2021 (n = 140).**

| Variable | Category | Frequency | Percentage |
|---|---|---|---|
| Reasons for not took AMT | Lack of information about the campaign | 37 | 26.40 |
| | Serious side effect during the previous campaign | 32 | 23 |
| | Not interested to mentioned their reasons | 20 | 14.30 |
| | Absence during the campaign | 16 | 11.40 |
| | Illness with other diseases during the campaign | 15 | 10.70 |
| | Fear of side effect | 10 | 7.10 |
| | Being Pregnant | 7 | 5 |
| | Having healthy eyes | 3 | 2.10 |

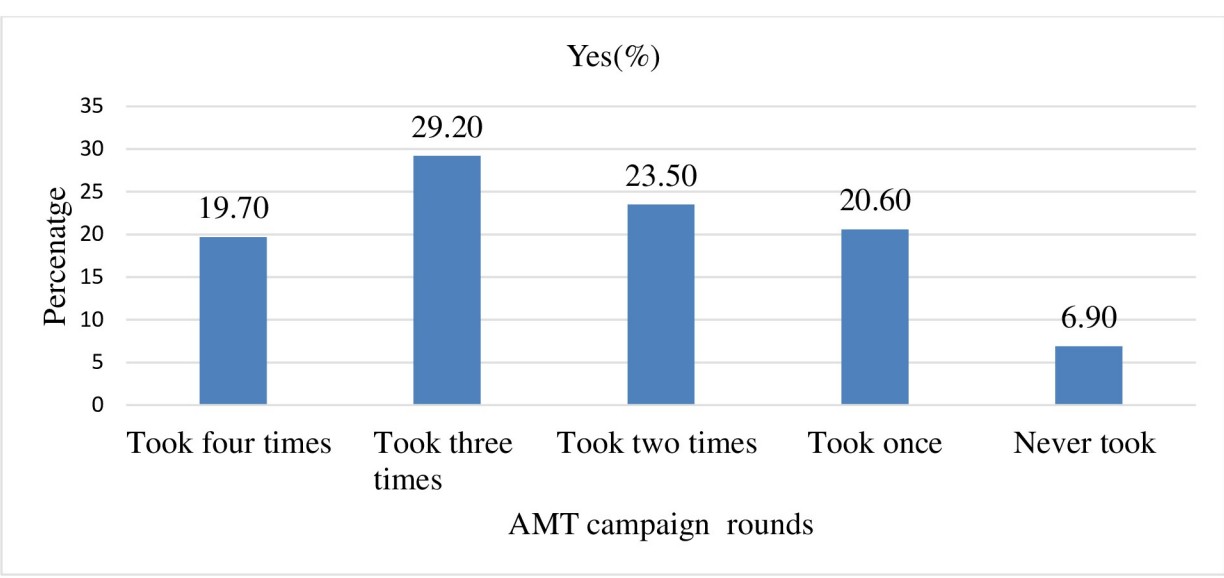

**Fig 2. Number of rounds respondents participated in past azithromycin mass treatment campaign in Goro district, Bale zone, Southeast Ethiopia, 2021 (n = 578).**

## AMT campaign timing and social mobilization

The majority, 337(62.30%) of the respondents responded that the AMT campaign was given at the quietest time (off-season harvesting and planting time). Accordingly, 204 (37.70%) respondents responded that the campaign was not given at the quiet time (overlap at the time of

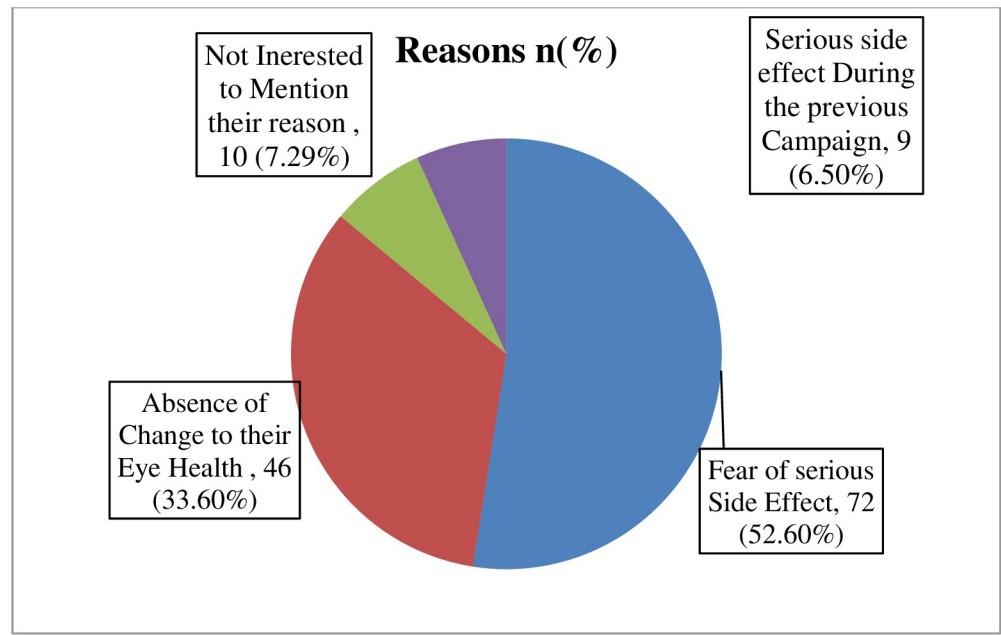

**Fig 3. Reasons for withdrawal of respondents from the future azithromycin mass treatment campaign in Goro district, Bale zone, Southeast Ethiopia, 2021 (n = 137).**

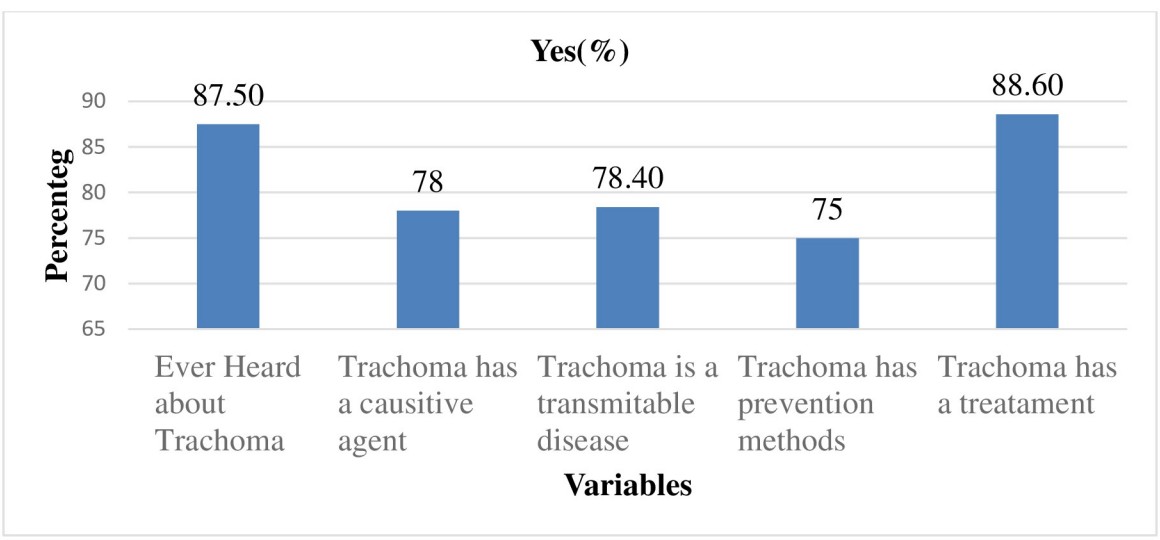

**Fig 4. Response to question related to knowledge about trachoma in Goro district, Bale zone, Southeast Ethiopia 2021 (n = 578).**

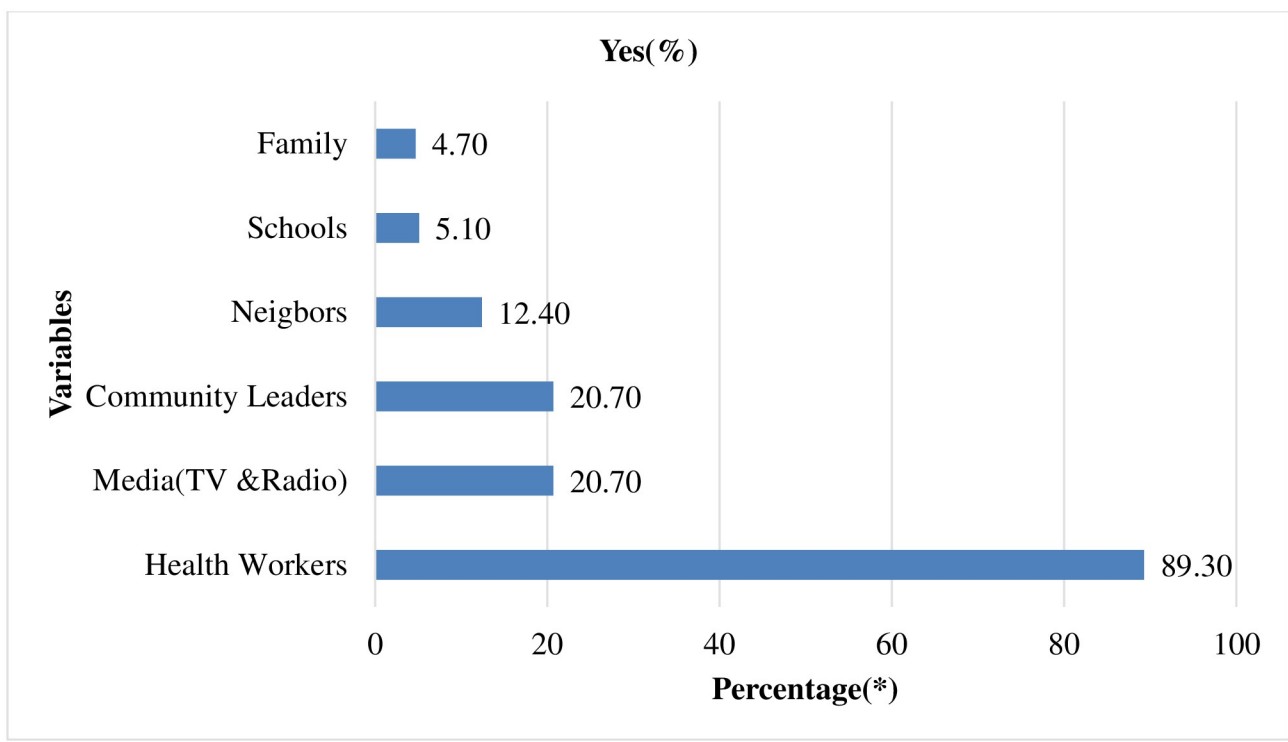

**Fig 5. Response to question related to the source of information about azithromycin mass treatment in Goro district, Bale zone, Southeast Ethiopia, 2021 (n = 507).**

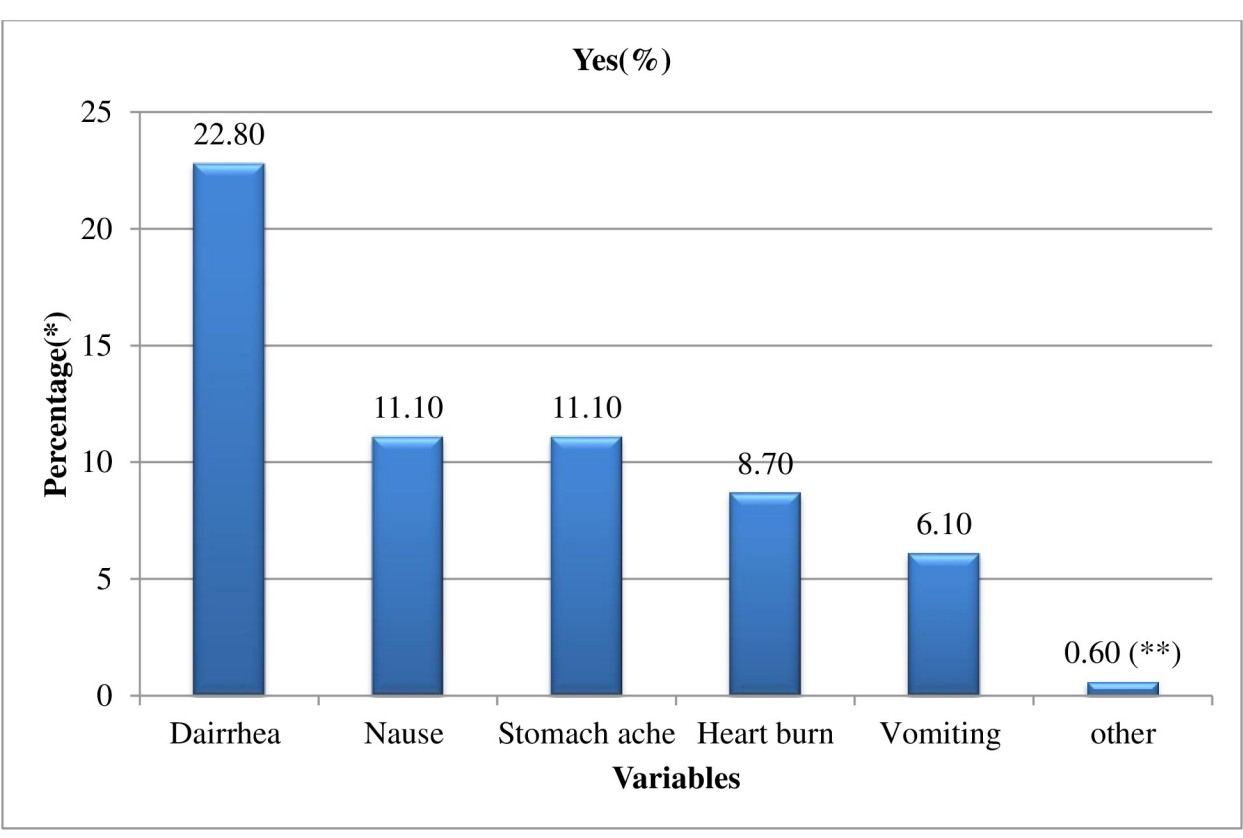

**Fig 6. Azithromycin mass treatment related side effects experienced by the respondents in Goro District, Bale zone, Southeast Ethiopia, 2021 (n = 167).** * Multiple response were possible ** Other (Headache, back pain).

harvesting and seeding time). More than half, 305(52.80%) of respondents heard about the campaign at least a day before the campaign date.

## Factors associated with coverage of azithromycin mass treatment

Educational status was associated with coverage of AMT. Respondents who had completed college and above were significantly associated with coverage of AMT (AOR = 7.20; 95% CI: 1.02–51.09). Those who had good knowledge about trachoma had a positive impact in complying with the AMT uptake (AOR = 2.36; 95% CI: 1.19–4.70), as well, respondents who had good knowledge about the azithromycin mass treatment were significantly associated with the coverage of AMT (AOR = 4.19; 95% CI: 2.19–7.98).

Being heard of serious adverse effects of azithromycin from other persons was also significantly associated with the coverage of AMT (AOR = 0.25; 95% CI: 0.10–0.59). Conducting AMT campaigns at the quietest times (off-harvesting/planting seasons) was significantly associated with the coverage of AMT (AOR = 6.23; 95% CI: 3.23–11.98). Being volunteered to take azithromycin in the future campaign was significantly associated with the coverage of azithromycin mass treatment (AOR = 5.46; 95% CI: 2.76–10.79) (Table 3).

## Discussion

This study assessed the coverage of azithromycin mass treatment for trachoma and associated factors among adults in the Goro district, Bale zone, Southeast Ethiopia. Having better

**Table 3. Factors associated with the coverage of AMT in Goro District, Bale zone, Southeast Ethiopia, 2021 (n = 578).**

| Variables | Uptake of AMT | | 95% Confidence interval | | | |
|---|---|---|---|---|---|---|
| | Yes (%) | No (%) | COR | p-value | AOR | p-value |
| **Sex** | | | | | | |
| Male | 200(69.40) | 88(30.60) | 1.00 | | 1.00 | |
| Female | 238(82.10) | 52(17.90) | 2(1.36, 2.97) | 0.001* | 1.72(0.93, 3.17) | 0.079 |
| **Educational status** | | | | | | |
| No -Formal education | 219(76.30) | 68(23.70) | 1.00 | | 1.00 | |
| Primary education | 131(70.10) | 56(29.90) | 0.72(0.48, 1.10) | 0.13 | 0.72(0.31, 1.68) | 0.460 |
| Secondary education | 39(73.60) | 14(26.40) | 0.86(0.44, 1.68) | 0.67 | 0.90(0.26, 3.17) | 0.880 |
| College & above | 49(96.10) | 2(3.90) | 7.60(1.80, 32.10) | 0.006* | 7.2(1.01, 51.09) | 0.047** |
| **Knowledge about Trachoma** | | | | | | |
| Good Knowledge | 357(79.90) | 90(20.10) | 2.44(1.60, 3.73) | 0.001* | 2.36(1.19, 4.70) | 0.015** |
| Poor knowledge | 81(61.80) | 50(38.20) | 1.00 | | 1.00 | |
| **Knowledge about AMT** | | | | | | |
| Good Knowledge | 280(85.60) | 47(14.40) | 3.50(2.34, 5.23) | 0.001* | 4.19(2.19, 7.98) | 0.001** |
| Poor Knowledge | 158(62.90) | 93(37.10) | 1.00 | | 1.00 | |
| **AMT campaign given at Quite time (n = 541)** | | | | | | |
| Yes | 301(89.30) | 36(10.70) | 5.17(3.31, 8.08) | 0.001* | 6.23(3.23, 11.98) | 0.001** |
| No | 126(61.80) | 78(38.20) | 1.00 | | 1.00 | |
| **Experience Side effect of the drug in the previous campaign** | | | | | | |
| Yes | 112(67.10) | 55(32.90) | 1.00 | | 1.00 | |
| No | 326(79.30) | 85(20.70) | 0.53(0.35, 0.79) | 0.002* | 0.72(0.33, 1.56) | 0.410 |
| **Heard Serious Adverse Effects of the drug from other person** | | | | | | |
| Yes | 29(37.70) | 48(62.30) | 1.00 | | 1.00 | |
| No | 409(81.60) | 92(18.40) | 0.13(0.08, 0.22) | 0.001* | 0.25(0.10, 0.59) | 0.002** |
| **Willing to take AMT in the future** | | | | | | |
| Yes | 385(87.30) | 56(12.70) | 10.89(6.99, 16.98) | 0.001* | 5.46(2.76, 10.79) | 0.001** |
| No | 53(38.70) | 84(61.30) | 1.00 | | 1.00 | |

AOR: adjusted odds ratio; COR: crude odds ratio;

*statistically significant in binary logistic regression analysis;

**statistically significant in multivariable logistic regression analysis.

knowledge about trachoma and AMT, being educated, AMT campaigns being conducted at a quiet time, hearing serious adverse effects from other persons, and being willing to take AMT in the future campaigns were significantly associated with the coverage of azithromycin mass treatment.

This study indicated that the proportion of azithromycin mass treatment coverage was 75.80%, which is below the WHO minimum coverage target of 80% [10]. The finding of this study was comparable with the 76.80% taken from a population-based coverage survey conducted in the Amhara region, Ethiopia [19]; 78.50% in Wogidie district, East of the Amhara region, Ethiopia [20], 75.40% in Didiba of Enderta district of Tigray region, Ethiopia [17], and 73% in the study conducted on the evaluation of a single dose of azithromycin for trachoma in low-prevalence communities [21]. The finding of this study was higher than 67.80% of the study conducted in the West Gojjam zone of the Amhara region, Ethiopia [19], and 60.3% in Nigeria [22]. However, the finding of this study was lower than 83% of the study conducted in six districts of the Oromia region, Western Ethiopia [13], 88.80% of the study conducted in

Goncha Siso Enderta woreda in the Amhara region, Ethiopia [23]) and 85.5% of the finding taken from 48 eligible communities in four Gambian districts [24]. The discrepancy could be due to the differences in social mobilization activities at different sites and study periods, as well as differences in the socio-cultural background of the respondents.

The study conducted on Targeting Antibiotics to Households for Trachoma Control among individuals from two communities in The Gambia (West Africa) and two communities in Tanzania (East Africa) indicated that treatment of a few persons in endemic areas results in reinfection from familial or neighborhood sources unless the treatment is more widespread [25]. This might hurt program success as elimination of ocular chlamydial infection is only achieved as long as the azithromycin mass treatment is given annually at a high enough coverage as per the WHO minimum target set of 80% [8].

The findings of this study revealed that respondents who completed college and above were seven times more likely to take azithromycin mass treatment than those who had no formal education. One possible explanation could be that educated respondents might have access to information regarding the disease and AMT and simply understand the message provided.

Respondents who had better knowledge about trachoma had a positive impact on their ability to comply with the azithromycin mass treatment uptake. Those who had better knowledge about trachoma were two times more likely to take azithromycin compared to their counterparts. This is supported by the study done in the Eastern Amhara region where knowledge of trachoma is significantly associated with AMT uptake [26]. A possible explanation could be that a generalized understanding of the causes, symptoms, and consequences of a disease might strongly motivate azithromycin mass treatment uptake. Respondents who had better knowledge about the azithromycin mass treatment were four times more likely to take azithromycin than those who had worse knowledge about AMT. This is agreed with the study conducted in four districts of the Amhara region, which indicated that having knowledge about the mass drug administration campaign makes people three times more likely to take AMT than their counterparts [20]), the study conducted in Awi zone, Amhara region which showed that provider message about AMT campaign increases the uptake of azithromycin and reduces fear of side effects [12], and effective social mobilization activities about the campaign is essential for successful program implementation [27]). Furthermore, this finding is consistent with the study conducted by Amarillo and his colleagues that health education and promotion activities, utilizing locally translated messages, education and communication materials, and other media, may increase the community's level of awareness and knowledge on the campaign [28]. Additionally, this result was in line with the WHO report on Global Trachoma Elimination, which said that social mobilization is a crucial element and to strengthen efforts, involving the correct stakeholders at the community level before starting the campaign is essential [29]. The possible explanation might be that enhancing the respondent's understanding of the benefits and side effects of the drug could, result in increased coverage of mass treatment.

Respondents who said that the campaign was given at a quiet time (off-season) were six times more likely to take azithromycin compared to their counterparts. This is in agreement with the study conducted on coverage, social mobilization, and challenges of the mass zithromax administration campaign in the South and Southeast zones of Tigray, Northern Ethiopia [17]. This might be most likely during the harvesting or planting time when people are not present at the home, and they had no time to take azithromycin mass treatment.

Respondents who heard about the serious adverse effects of the drug from another person were 75% less likely to take azithromycin mass treatment than their counterparts. This is supported by the similar findings of the study conducted in Amhara and Tigray regional state, which indicated that fear of side effects was one of the mentioned reasons for not swallowing

the drug [12,17]. Another study conducted in Kenya also agreed with this study and showed that fear of side effects was reported as barriers to swallowing the drug [30]), and another study conducted on neglected tropical diseases of mass drug administration; including trachoma, showed that one reason for not-took mass drug administration is fear of side effects [31]. The possible explanation might be that of rumors about the side effects of the drug have spread fast in the community rather than its benefit if not managed accordingly.

Respondents who volunteered to take the azithromycin mass treatment in the future were five times more likely to comply with the azithromycin mass treatment uptake as compared to their counterparts. This finding is consistent with the study conducted in Awi zone, Amhara region that showed those who expressed their willingness to take azithromycin in the future campaign were better taken in the last AMT campaign (12). A possible explanation might be that those who understand the general benefits and side effects of AMT had an interest to take azithromycin mass treatment in a future campaign.

One limitation of this study is the use of self-reporting methods to assess AMT uptake, which might lead to under or over-reporting.

## Conclusion

The proportion of azithromycin mass treatment (AMT) coverage in Goro district found to be lower than the WHO minimum target. Having better knowledge about the trachoma and AMT and being educated were some of the factors positively affecting the uptake of the azithromycin mass treatment. Hearing serious adverse effects of the drug from other person and campaigns conducted during harvesting or seeding time were some factors negatively affecting the coverage of AMT in the current study. These findings suggested the need to strengthen social mobilization about the AMT campaign, as well as due consideration of additional campaigns during the quiet time (off-harvesting/planting seasons); so that trachoma elimination is achieved as long as the azithromycin mass treatment is given annually at a high enough coverage. Besides, the future studies better explore what would it take for the individuals to participate in the subsequent rounds of AMT campaigns.

## Supporting information

**S1 Fig. Schematic presentation of sampling procedure to assess azithromycin mass drug administration uptake and associated factors in Goro district, Bale Zone, Southeast Ethiopia, 2021.**
(DOCX)

## Acknowledgments

We would like to thank study participants for their willingness to participate. We also would like to extend our appreciation to the people working at district health office for their supports.

## Author Contributions

**Conceptualization:** Tadele Feyisa, Desalegn Bekele, Birhanu Tura, Ahmednur Adem, Fikadu Nugusu.

**Data curation:** Tadele Feyisa, Desalegn Bekele, Birhanu Tura, Ahmednur Adem, Fikadu Nugusu.

**Formal analysis:** Tadele Feyisa, Desalegn Bekele, Birhanu Tura, Ahmednur Adem, Fikadu Nugusu.

**Funding acquisition:** Tadele Feyisa.

**Investigation:** Tadele Feyisa, Ahmednur Adem, Fikadu Nugusu.

**Methodology:** Tadele Feyisa, Desalegn Bekele, Birhanu Tura, Ahmednur Adem, Fikadu Nugusu.

**Project administration:** Tadele Feyisa, Birhanu Tura, Ahmednur Adem, Fikadu Nugusu.

**Resources:** Tadele Feyisa.

**Software:** Tadele Feyisa, Desalegn Bekele, Birhanu Tura, Ahmednur Adem, Fikadu Nugusu.

**Supervision:** Tadele Feyisa, Desalegn Bekele, Birhanu Tura, Ahmednur Adem, Fikadu Nugusu.

**Validation:** Tadele Feyisa, Desalegn Bekele, Birhanu Tura, Ahmednur Adem, Fikadu Nugusu.

**Visualization:** Tadele Feyisa, Desalegn Bekele, Birhanu Tura, Ahmednur Adem, Fikadu Nugusu.

**Writing – original draft:** Tadele Feyisa, Desalegn Bekele, Birhanu Tura, Ahmednur Adem, Fikadu Nugusu.

**Writing – review & editing:** Tadele Feyisa, Birhanu Tura, Ahmednur Adem, Fikadu Nugusu.

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
