## [Decision Letter · Decision Letter 0]

21 Feb 2022

Dear Mr Nugusu,

Thank you very much for submitting your manuscript "To eliminate trachoma: azithromycin mass drug administration coverage and associated factors among adults in Goro district, southeast Ethiopia" for consideration at PLOS Neglected Tropical Diseases. As with all papers reviewed by the journal, your manuscript was reviewed by members of the editorial board and by several independent reviewers. In light of the reviews (below this email), we would like to invite the resubmission of a significantly-revised version that takes into account the reviewers' comments. 

We cannot make any decision about publication until we have seen the revised manuscript and your response to the reviewers' comments. Your revised manuscript is also likely to be sent to reviewers for further evaluation.

Sincerely,

Alberto Novaes Ramos Jr

Associate Editor

Andrés Henao-Martínez

Deputy Editor

Reviewer's Responses to Questions

**Key Review Criteria Required for Acceptance?**

**Methods**

-Are the objectives of the study clearly articulated with a clear testable hypothesis stated?

-Is the study design appropriate to address the stated objectives?

-Is the population clearly described and appropriate for the hypothesis being tested?

-Is the sample size sufficient to ensure adequate power to address the hypothesis being tested?

-Were correct statistical analysis used to support conclusions?

-Are there concerns about ethical or regulatory requirements being met?

Reviewer #1: The methods presented here are straight forward and generally appropriate for the data. I do however have a few concerns and suggestions:

1) If the analysis wasn’t weighted, then there is a bias in the results favoring responses from individuals living in households with few adults because the team randomly selected only one adult per household. If the study team collected data on the number of adults in each household then they can weight the analysis appropriately (by using 1 / [number of adults in the HH] as the weight). If this cannot be done, then it is important to be honest about this bias. If household size is associated with coverage then the findings are not representative. 

2) Knowledge was deemed to be ‘good’ if the respondent scored points higher than the mean, but this is not a sign of good knowledge but rather a sign of knowing more than most people around you. If the entire population has low knowledge, scoring above the mean is not a sign that one has good knowledge. It would make more sense to me to set a threshold above which anyone scoring would be considered to have good knowledge (e.g. if you answer X or more of the questions correctly). Alternatively, the team could leave the analysis as it has been done and replace good vs. bad knowledge with ‘better’ vs. ‘worse’ – this is more correct because it suggests that knowledge is being assessed relatively between two groups. 

3) Did the analysis account for clustering at the Kebele level? Since a design effect was included in the sample size calculations, it suggests that the analysis intends to account for correlation within clusters. It is important to specify if and how this was done, as many previous studies have shown high correlation within clusters when it comes to mass treatment coverage.

Reviewer #2: Source and study population, inclusion criteria as well as exclusion criteria were not stated.

Reviewer #3: The authors conducted a community-based cross-sectional study using a structured interviewer-administered questionnaire.

The objectives of the study were clearly articulated, with an appropriate study design.

They clearly described how the sample size was estimated, and the statistical analysis used to support conclusions.

There are no concerns about ethical or regulatory requirements in the study.

**Results**

-Does the analysis presented match the analysis plan?

-Are the results clearly and completely presented?

-Are the figures (Tables, Images) of sufficient quality for clarity?

Reviewer #1: I would like to suggest the following revisions:

1) It is important to be consistent with the use of significant figures (ie, how many numbers are reported after the decimal) in all the numbers (CIs, %s, means, etc.) that are reported in the tables and text. 

2) Table 1 – formatting is off, some of the variable names in Column 1 appear to be centered in the cell while others appear to be at the top of the row, which makes for difficult reading. 

3)Figure 3 – Please add the counts to this pie chart and correct the typos in the boxes.

4)I don’t see the crude results for ‘conducting campaigns at the quietest time’; please include this information in Table 1 or one of the later tables.

Reviewer #2: The results should be stated in line to the objectives

Reviewer #3: The analysis presented matched the analysis plan.

The results were clearly and completely presented.

The figures (Tables, Images) are of sufficient quality for clarity.

**Conclusions**

-Are the conclusions supported by the data presented?

-Are the limitations of analysis clearly described?

-Do the authors discuss how these data can be helpful to advance our understanding of the topic under study?

-Is public health relevance addressed?

Reviewer #1: The authors do not adequately describe some key limitations of the study and there is room for greater discussion on the implications of these findings. I would like to suggest that they address the following in the discussion:

1) How complete is the Community Health Information System, in terms of its inclusion of all households? Is this list kept up-to-date locally? Is there a risk that some new households, ethnic minorities, or other marginal groups might be missing?

2) One potential bias of this study is that the respondents were limited to people who were home at the time of the survey team’s visit. How might this impact your results? I suggest you comment on this in the discussion. 

3)Line 358 – I don’t understand this statement. The current study did not indicate anything about reinfection. If the authors are referring to reference 26, then then need to be more clear in the opening of the sentence that they are referring to another study. 

4)The fact that willingness to comply with treatment in future rounds was so significant is potentially worrisome to trachoma programs. It suggests that there may be strong systematic compliance (and conversely systematic non-compliance) whereby the same group in the population is routinely participating while others are not. It would be good to understand what would be required to get the non-participants to participate in the future. Perhaps adding the question, “what would it take for you to participate in future rounds of AMT?” would be beneficial to add to a future study. 

5) The discussion would be strengthened if, rather than just point out all the places where this study’s findings agree with past studies, the authors also added in implications for the trachoma program and recommendations for how to improve implementation of AMT and achieve higher coverage.

Reviewer #2: How the current coverage of the AMT 75% inline to WHO recommendation (80%)?

Reviewer #3: The conclusions were supported by the data presented.

The authors highlighted the need to strengthen social mobilization about the Azithromycin Mass Treatment campaign and due consideration of additional campaigns at the non-harvesting nor the planting time for other trachoma endemic areas.

**Editorial and Data Presentation Modifications?**

Reviewer #1: This study needs a very close proof read by a native English speaker. There are too many grammatical errors throughout the paper for me to comment on here. Many have to do with the use of prepositions and noun agreement.

Reviewer #2: Abstract

The background lacks aim of the study

methods: 

Line 37: Principal finding should be expunged. 

Results

Which one is appropriate for your study? prevalence or magnitude? 

conclusion

Line 47: Significance should also be expunged. 

How the current finding 75% in line to WHO recommendation (80%)

Keywords: Add coverage

Main body

Introduction

intervention measures to mitigate the problems should be stated after the rationale of the study were indicated.

Methods

Line 122: Rearrange it

Source and study population, inclusion criteria as well exclusion criteria were not specified.

Outcome assessment should be referenced.

Why did the authors use mean for knowledge question assessment?

Have you checked intervention of variables and identify confounding factors?

State why did the authors use multivariable LR analysis?

Restate line 215-216.

Line 217- 218 should be written after line 217.

State how did you assess normality of the data distribution?

ethical consideration

State tenet of Helsinki Declaration

Results

Line 227: Restate as ' Sociodemographic and economic characteristics of ....

Substitute majority by most through out the results part

Line 34-316: should be expunged as they are not the objective of the study.

Line 262: should be merged with table 2.

Discussion

The discussion is clumsy. 

Line 345: Did the CI overlap each other to use the word "comparable"? 

Limitation of the study was not indicated

conclusion

As commented in the abstract section

Reviewer #3: Minor comment: correct the figure number in the legends for figures 2-4.

Rephrase the last sentence, starting from line 425, to clarify the confusing language.

**Summary and General Comments**

Reviewer #1: This article presents the results of a coverage survey for azithromycin mass treatment in Ethiopia. I would like to start by applauding the study's authors for undertaking this work. Coverage surveys are an important monitoring tool to help NTD programs understand how the implementation of the program and may be used to identify important gaps in treatment and areas for improvement. Unfortunately, coverage surveys are not routinely conducted by most trachoma programs and thus this study is a good one to add to the evidence base. 

The team appears to have done a diligent job making sure the questionnaire was well-understood, appropriate, and completed for each respondent. I appreciate the inclusion of a dedicated pre-test and refinement period. 

It is very interesting, and perhaps reassuring, that ‘Azithromycin not offered’ or ‘Nobody came’ were not among the reasons mentioned for not taking the drug. This suggests that the majority of the challenges with coverage in this district are related to the participants (both their physical presence and psychological reasoning) and not the distributors or supply chain.

Reviewer #2: The aim of the study should be specified in the abstract.

Source and study population, inclusion criteria as well exclusion criteria were should be specified.

Outcome assessment should be cited.

Why did the authors use mean for knowledge question assessment?

Reviewer #3: The study was well designed, analyzed and interpreted.

PLOS authors have the option to publish the peer review history of their article (what does this mean?). If published, this will include your full peer review and any attached files.

Reviewer #1: No

Reviewer #2: Yes: Girma Beressa

Reviewer #3: Yes: Taher Eleiwa
---

## [Decision Letter · Decision Letter 1]

20 May 2022

Dear Mr Nugusu,

We are pleased to inform you that your manuscript 'To eliminate trachoma: azithromycin mass drug administration coverage and associated factors among adults in Goro district, southeast Ethiopia' has been provisionally accepted for publication in PLOS Neglected Tropical Diseases.

Best regards,

Alberto Novaes Ramos Jr

Associate Editor

Andrés Henao-Martínez

Deputy Editor

Reviewer's Responses to Questions

**Key Review Criteria Required for Acceptance?**

**Methods**

-Are the objectives of the study clearly articulated with a clear testable hypothesis stated?

-Is the study design appropriate to address the stated objectives?

-Is the population clearly described and appropriate for the hypothesis being tested?

-Is the sample size sufficient to ensure adequate power to address the hypothesis being tested?

-Were correct statistical analysis used to support conclusions?

-Are there concerns about ethical or regulatory requirements being met?

Reviewer #2: The source and study population, as well as inclusion criteria, should be succinctly delineated.

Reviewer #3: The authors had all the concerns addressed.

**Results**

-Does the analysis presented match the analysis plan?

-Are the results clearly and completely presented?

-Are the figures (Tables, Images) of sufficient quality for clarity?

Reviewer #2: Yes

Reviewer #3: The analysis and the results are concordant and clearly articulated. The figures are clear.

**Conclusions**

-Are the conclusions supported by the data presented?

-Are the limitations of analysis clearly described?

-Do the authors discuss how these data can be helpful to advance our understanding of the topic under study?

-Is public health relevance addressed?

Reviewer #2: Yes

Reviewer #3: The conclusions were supported by the data presented.

**Editorial and Data Presentation Modifications?**

Reviewer #2: Some technical problems (technical, punctuation, and capitalization should be maintained) throughout the entire manuscript.

Reviewer #3: (No Response)

**Summary and General Comments**

Reviewer #2: Is the word ' magnitude appropriate in this study?

The authors should indicate how they assessed normality distribution of the data.

Reviewer #3: (No Response)

PLOS authors have the option to publish the peer review history of their article (what does this mean?). If published, this will include your full peer review and any attached files.

Reviewer #2: **Yes: **Girma Beressa

Reviewer #3: **Yes: **Taher Eleiwa

---

## [Editor Report · Acceptance letter]

10 Jun 2022

Dear Mr Nugusu,

We are delighted to inform you that your manuscript, "To eliminate trachoma: azithromycin mass drug administration coverage and associated factors among adults in Goro district, southeast Ethiopia," has been formally accepted for publication in PLOS Neglected Tropical Diseases.

Best regards,

Shaden Kamhawi

co-Editor-in-Chief

Paul Brindley

co-Editor-in-Chief
